# The Antiviral and Virucidal Activities of Voacangine and Structural Analogs Extracted from *Tabernaemontana cymosa* Depend on the Dengue Virus Strain

**DOI:** 10.3390/plants10071280

**Published:** 2021-06-23

**Authors:** Laura Milena Monsalve-Escudero, Vanessa Loaiza-Cano, Maria Isabel Zapata-Cardona, Diana Carolina Quintero-Gil, Estiven Hernández-Mira, Yina Pájaro-González, Andrés Felipe Oliveros-Díaz, Fredyc Diaz-Castillo, Wistón Quiñones, Sara Robledo, Marlen Martinez-Gutierrez

**Affiliations:** 1Grupo de Investigación en Ciencias Animales-GRICA, Facultad de Medicina Veterinaria y Zootecnia, Universidad Cooperativa de Colombia, Bucaramanga 680005, Colombia; lauramilemo@hotmail.com (L.M.M.-E.); vanessa.loaiza@udea.edu.co (V.L.-C.); mariaisab5@gmail.com (M.I.Z.-C.); dcaro63@gmail.com (D.C.Q.-G.); estiven_hdz@hotmail.com (E.H.-M.); 2Laboratorio de Investigaciones Fitoquímicas y Farmacológicas de la Universidad de Cartagena—LIFFUC, Universidad de Cartagena, Cartagena 130001, Colombia; yinapajaro@mail.uniatlantico.edu.co (Y.P.-G.); aoliverosd@unicartagena.edu.co (A.F.O.-D.); fdiazc1@unicartagena.edu.co (F.D.-C.); 3Grupo de Investigación en Farmacia Asistencial y Farmacología, Universidad del Atlántico, Barranquilla 080001, Colombia; 4Grupo de Química Orgánica de Productos Naturales, Universidad de Antioquia, Medellín 050001, Colombia; winston.quinones@udea.edu.co; 5Programa de Estudio y Control de Enfermedades Tropicales-PECET, Universidad de Antioquia, Medellín 050001, Colombia; sara.robledo@udea.edu.co

**Keywords:** dengue virus, *Tabernaemontana cymosa*, indole alkaloids, molecular docking, antivirals

## Abstract

Currently, no specific licensed antiviral exists for treating the illness caused by dengue virus (DENV). Therefore, the search for compounds of natural origin with antiviral activity is an important area of research. In the present study, three compounds were isolated and identified from seeds of *Tabernaemontana cymosa* plants. The in vitro antiviral effect of those compounds and voacangine against different DENV strains was assessed using different experimental approaches: compounds added before the infection (Pre), at the same time with the virus (Trans), after the infection (Post) or compounds present in all moments of the experiment (Pre-Trans-Post, Combined treatment). In silico studies (docking and molecular dynamics) were also performed to explain the possible antiviral mechanisms. The identified compounds were three structural analogs of voacangine (voacangine-7-hydroxyindolenine, rupicoline and 3-oxo-voacangine). In the Pre-treatment, only voacangine-7-hydroxyindolenine and rupicoline inhibited the infection caused by the DENV-2/NG strain (16.4% and 29.6% infection, respectively). In the Trans-treatment approach, voacangine, voacangine-7-hydroxyindolenine and rupicoline inhibited the infection in both DENV-2/NG (11.2%, 80.4% and 75.7% infection, respectively) and DENV-2/16681 infection models (73.7%, 74.0% and 75.3% infection, respectively). The latter strain was also inhibited by 3-oxo-voacangine (82.8% infection). Moreover, voacangine (most effective virucidal agent) was also effective against one strain of DENV-1 (DENV-1/WestPac/74) and against the third strain of DENV-2 (DENV-2/S16803) (48.5% and 32.4% infection, respectively). Conversely, no inhibition was observed in the post-treatment approach. The last approach (combined) showed that voacangine, voacangine-7-hydroxyindolenine and rupicoline inhibited over 90% of infections (3.5%, 6.9% and 3.5% infection, respectively) of both strains (DENV-2/NG and DENV-2/16681). The free energy of binding obtained with an in silico approach was favorable for the E protein and compounds, which ranged between −5.1 and −6.3 kcal/mol. Finally, the complex formed between DENV-2 E protein and the best virucidal compound was stable for 50 ns. Our results show that the antiviral effect of indole alkaloids derived from *T. cymose* depends on the serotype and the virus strain.

## 1. Introduction

Among the infections caused by viruses transmitted by arthropods, dengue fever is the most frequent, with approximately 50–100 million cases worldwide each year [1]. This illness is caused by the dengue virus (DENV). Approximately 2500 million people inhabit endemic areas that host about 120 million visitors every year. Therefore, the global incidence of DENV and, consequently, the sickness that it causes has increased during the last 40 years, particularly in Latin America [1]. During the 1960s and early 1970s, DENV transmission was partly interrupted in the American continent as a result of a campaign to eradicate the transmitting mosquito (*Aedes aegypti*) [2]. However, vector surveillance and control measures were discontinued, and mosquitoes reappeared in large quantities, which in turn caused outbreaks in Latin America in the late 1970s and early 1980s [3]. Colombia is considered a hyperendemic country where the four serotypes of the virus cocirculate, and five important epidemics have occurred in the last 50 years. The highest number of cases were reported in 2010, 2019, and 2013, with the outbreak in 2013 showing the highest number of cases of severe dengue fever [4].

DENV is a virus belonging to the genus *Flavivirus* of the family *Flaviviridae*. The viral particle comprises a lipid envelope that anchors the E protein and pr/M protein, depending on its maturation stage, and a single-stranded, positive-sense RNA of approximately 11 kb protected by a nucleocapsid. This genome has a single open reading frame, which codes a polyprotein that in turn, produces three structural proteins (Capsid (C), Precursor of the Membrane Protein/Membrane Protein (prM/M), and Envelope (E)) and seven non-structural proteins (NS1, NS2a, NS2b, NS3, NS4a, NS4b and NS5) that are essential for the productive infection [5]. The E protein is involved in the early steps of viral replication (adhesion and fusion) and additionally, its structure carries the antigenic determinants that define the four serotypes of the virus (DENV-1, DENV-2, DENV-3 and DENV-4). These are further classified into genotypes, and finally, into strains. The virus strains belonging to serotype 2 include the New Guinea strain (DENV-2/NG), isolated in that region from a patient with dengue fever in 1994, and the 16,681 strain (DENV-2/16681), isolated in Thailand from a patient with severe dengue fever in 1964 [6]. The two strains show some structural differences that result in different pathogeneses [7] and have, therefore, been associated with the development of dengue fever or severe dengue fever, respectively. For this reason, they have been widely used in in vitro studies that have evinced differential effects on the expression of cell proteins [8,9].

The DENV replication cycle starts with the attachment of the viral E protein to host membrane receptors, which leads to endocytosis mediated by diverse routes [10]. The acidification of the endocytic vesicles causes a conformational change of the E protein, which allows the fusion of the endosomal membrane with the viral envelope and the release of the virus nucleocapsid into the cytoplasm [11]. The protein translation in the ribosomes on the endoplasmic reticulum produces a complete polypeptide. This is modified and later gives rise to the viral proteins involved in the generation of negative-sense intermediary RNA strands that act as templates for new positive-sense strands [4]. Subsequently, the nucleocapsid interacts with the new RNA molecules, and the viral particles are assembled. The viral maturation process involves the association of prM/M and E proteins before the occurrence of post-translational modifications, producing mature M protein. The final step is the exocytosis to the extracytoplasmic space, where the neutral pH induces the cleavage of the pr peptide. This, in turn, produces the final conformation of the E protein that can now be recognized again by receptors of another cell [12]. All the stages of the viral replication cycle are susceptible to inhibition [13], but despite the considerable number of studies including research, development and assessment of antivirals conducted to date, there is not a single licensed drug intended for use in infected patients. Therefore, studies in this area continue to be of great interest.

Biodiversity has made the study of plants an important source for the possible isolation and characterization of molecules, whose potential against causal agents of disease, including viruses [14], can be initially assessed in vitro. In this regard, pharmacognosy has enabled the study of active principles or molecules obtained from natural products with therapeutic potential, resulting in their application in the pharmaceutical industry [15]. Our research group has demonstrated the anti-arboviral potential of ethanol extracts from plants in the Caribbean region of Colombia [16] and of some compounds derived from them, including *Mammea americana*, *Psidium guajava*, and *Tabernaemontana cymosa* [17,18].

The family Apocynaceae is one of the largest families of angiosperms. This family currently comprises over 300 genera distributed in the subfamilies Apocynoideae, Asclepiadoideae, Periplocoideae, Secamonoideae, and Rauvolfioideae. The genus *Tabernaemontana* [19] belongs to the latter subfamily and is known to produce iboga-type indole alkaloids [20]. The genus *Tabernaemontana*, one of the largest groups in the subfamily Rauvolfioideae, includes approximately 100 species distributed in different tropical regions worldwide. This genus is widely used in ethnobotanical medicine because of its various documented biological activities, such as antioxidant [21], anti-inflammatory [22], anticarcinogenic [23] and antimicrobial activity [24]. In particular, larvicidal activity against larval stages III and IV of *Aedes aegypti* [25] and antiviral activity against various strains of DENV-2 and cell lines [16] have been observed in ethanol extracts from the species *Tabernaemontana cymosa* Jacq (*T. cymosa*). Moreover, two other compounds (lupeol acetate and voacangine (VOAC) isolated from this species showed inhibitory properties on DENV replication [17].

Considering the need to continue the search for specific antivirals against DENV and the previous findings on the antiviral potential of compounds extracted from *T. cymosa*, this study aimed to assess the antiviral effect of other compounds extracted from this plant on infections by various strains of DENV-2 and other serotypes, as well as to explore in silico interactions that may contribute to understanding the possible mechanism of antiviral action. Our results evidence that indole alkaloids derived from this plant are promising molecules for the development of anti-DENV drugs and that this effect depends on the serotype and the virus strain.

## 2. Results

### 2.1. Identification of the Compounds Derived from T. cymosa and Their Toxic Effect on Vero Cells

Three compounds were isolated from *T. cymosa*, as the VOAC previously reported [17]. The three reported alkaloids showed over 90% purity.

The compound Voacangine 7-hydroxyindolenine (VOAC-OH; TcK002) exhibited the following physical and spectral properties: Amorphous yellow-green solid.; R_f_ 0.67 (hexane: ethyl acetate 6:4); ^1^H NMR (300 MHz, CDCl_3_): δ 7.36 (d, J = 9 Hz, H-12), 6.91 (s, H-9), 6.80 (dd, J = 6 y 3 Hz, H-11), 3.82 (s, OCH_3_), 3.70 (s, CO_2_CH_3_), 0.86 (t, J = 7 Hz, H-18) ppm. ^13^C NMR (75 MHz, CDCl_3_): δ 186.97 (C-2), 174.05 (CO_2_Me), 159.23 (C-10), 144.90 (C-13), 144.53 (C-8), 121.49 (C-12), 113.85 (C-11), 108.09 (C-9), 88.43 (C-7), 58.68 (C-4), 55.88 (ArOCH_3_—C16), 53.39 (CO_2_CH_3_), 49.22 (C-5), 48.75 (C-3), 37.68 (C-20), 34.63 (C-17), 34.25 (C-6), 32.14 (C-15), 27.08 (C-14), 26.62 (C-19), 11.71 (C-18) (Figure 1A,D). The melting point was 135–137 °C. These spectra were compared with those reported in the literature to confirm the compound identification [26,27] (Appendix A).

The compound Rupicoline (TcK004) exhibited the following physical and spectral properties: yellow crystals; R_f_ 0.45 (chlorophorm: acetone 6:4); ^1^H NMR (300 MHz, CDCl_3_): δ 7.07 (dd, J = 8 y 3 Hz, H-11), 7.02 (s, H-9), 6.76 (d, J = 9 Hz, H-12), 4.29 (s, N-H), 3.95 (s, H-4), 3.76 (s, OCH_3_), 3.30 (s, CO_2_CH_3_), 1.91 (s, H-14), 0.91 (t, J = 6 Hz, H-18) ppm. ^13^C NMR (75 MHz, CDCl_3_): 202.9 (C-7), 174.57 (COOCH_3_/C-21), 154.21 (C-10), 153.82 (C-13), 126.89 (C-11), 121.78 (C-8), 114.12 (C-12), 104.63 (C-9), 55.90 (ArOCH_3_), 52.07 (C-3), 52.07 (C-16), 52.05 (C-4), 51.31 (CO_2_CH_3_), 47.67 (C-5), 35.81 (C-20), 31.1 (C-15), 30.8 (C-17), 28.69 (C-19), 26.08 (C-14), 25.77 (C-6), 12.14 (C-18) (Figure 1B,E). The melting point was 252.4–253.6 °C. These spectra were compared with those reported in the literature to confirm the compound identification [28] (Appendix A).

The compound 3-Oxo-Voacangine (OXO-VOAC; TcK005) exhibited the following spectral properties: ^1^H NMR (300 MHz, CDCl_3_): δ 7.92 (s, NH), 7.16 (d, J = 8.7 Hz, H-12), 6.95 (d, J = 2.1 Hz, H-9), 6.84 (dd, J = 8.7 y 2.4 Hz, H-11), 3.88 (s, OCH_3_), 3.77 (s, CO_2_CH_3_), 0.97 (t, J = 7 Hz, H-18) ppm. ^13^C NMR (75 MHz, CDCl_3_): 175.87 (C-3), 173.07 (CO_2_Me), 154.21 (C-10), 134.65 (C-2), 130.83 (C-13), 128.23 (C-8), 112.59 (C-11), 111.38 (C-12), 109.22 (C-9), 100.48 (C-7), 56.11 (C-16), 56.02 (C-4), 55.60 (OCH_3_), 53.09 (CO_2_CH_3_), 42.72 (C-5), 38.21 (C-20), 35.97 (C-14), 35.49 (C-17), 31.03 (C-15), 27.67 (C-19), 21.18 (C-6), 11.41 (C-18) (Figure 1C,F). The melting point was 253–254 °C. These spectra were compared with those reported in the literature to confirm the compound identification [27] (Appendix A).

Cell viability of the new VOAC-analog indole alkaloids was determined by the MTT method. As previously reported for VOAC [17], the viability was influenced differently depending on the compound and concentration evaluated. Viability was close to 80% in all cultures treated with the lowest concentration (8.1 µM in all cases). In contrast, viability decreased drastically when the highest concentrations were assessed (260.1 µM for VOAC-OH and rupicoline, and 261.5 µM for OXO-VOAC); furthermore, VOAC-OH was the compound that most affected viability (32.2% to 80.4%) (Figure 1D); meanwhile, the viability for rupicoline and 3-oxo-VOAC were higher than VOAC-OH and similar for both compounds in all evaluated concentrations (Figure 1E,F). According to these results, the working concentrations with viability values on Vero cells close to or greater than 80% were selected, 16.4 µM for VOAC-OH and rupicoline, and 16.5 µM for OXO-VOAC. The working concentration of VOAC was 17.1 µM [17].

### 2.2. Pre-Treatment with VOAC-OH and Rupicoline Inhibits the Infection by DENV-2/NG Strain

In the Vero cell cultures initially treated with the compounds and then infected with the DENV-2/NG strain, only VOAC-OH and rupicoline inhibited infection significantly (3.75 × 10^2^ PFU/mL, 16.4%; and 6.75 × 10^2^ PFU/mL, 29.6%, respectively, *p* < 0.05) compared with the control without compounds (2.28 × 10^3^ PFU/mL). None of the compounds inhibited the production of infectious viral particles in cultures infected with the DENV-2/16681 strain. Significant inhibition was observed for both virus strains when the positive inhibition control (heparin) was used (1.28 × 10^3^ PFU/mL, 56.0% DENV-2/NG; and 2.24 × 10^2^ PFU/mL, 55.5%, DENV-2/16681) (Figure 2).

### 2.3. The VOAC Virucidal Effect Depends on the Serotype of the Virus

When incubating the compounds with the DENV-2/NG strain before inoculating the cells, three out of four compounds showed significant inhibition of infectious viral particle production (VOAC 9.63 × 10^2^ PFU/mL, 11.2%; VOAC-OH 6.90 × 10^3^ PFU/mL, 80.4%; and rupicoline 6.50 × 10^3^ PFU/mL, 75.7%) compared with the control without any compounds (8.58 × 10^3^ PFU/mL). In cultures infected with the other virus strain (DENV-2/16681) directly exposed to the compounds, four compounds inhibited the infection significantly (VOAC 5.73 × 10^3^ PFU/mL, 73.7%; VOAC-OH 5.75 × 10^3^ PFU/mL, 74.0%; rupicoline 5.85 × 10^3^ PFU/mL, 75.3%; and OXO-VOAC 6.43 × 10^3^ PFU/mL, 82.8%) compared with the control without compounds (7.77 × 10^3^ PFU/mL). Statistically significant antiviral activity was observed for the positive inhibition control (heparin) against both strains (DENV-2/NG 3.59 × 10^3^ PFU/mL, 41.8%; and DENV-2/16681 3.18 × 10^3^ PFU/mL, 40.9%) (Figure 3).

Considering that VOAC presented the greatest inhibition in this approach, indicating its virucidal potential, this compound was evaluated against infection by other virus strains and serotypes. The results showed that VOAC had a virucidal effect against one DENV-1 strain (DENV-1/Westpac/74) and the third strain of DENV-2 (DENV-2/S16803), with 48.5% infection (1.31 × 10^9^ PFU/mL compared with 2.71 × 10^9^ PFU/mL in control without compounds) and 32.4% infection (*p* < 0.05) (2.38 × 10^10^ PFU/mL compared with 7.33 × 10^10^ PFU/mL in control without compounds), respectively. No virucidal activity was observed against the strains of serotypes 3 and 4 (Figure 4).

### 2.4. None of the Compounds Inhibit Post-Entry Stages of the Viral Reproductive Cycle

No statistically significant differences were observed between the treated and control (without compounds) cultures when cells were first infected with any of the two strains and then treated with any of the four compounds. Statistically significant antiviral activity was observed for the positive inhibition control (ribavirin) compared with the control without compounds for the DENV-2/NG strain (2.06 × 10^3^ PFU/mL, 56.2% vs. 3.67 × 10^3^ PFU/mL) as well as the DENV-2/16681 strain (8.17 × 10^2^ PFU/mL, 35.5% vs. 2.30 × 10^3^ PFU/mL) (Figure 5).

### 2.5. The Combined Antiviral Approach Using VOAC, VOAC-OH and Rupicoline Favors the Antiviral Effect Compared with Individual Approaches

In both viral models, significant inhibition of infection was observed when a given culture was subjected to Combined treatment (pre-trans-post-treatment) with the isolated alkaloids. Infection levels in the DENV-2/NG model were 3.5% (1.88 × 10^3^ PFU/mL), 6.9% (3.75 × 10^3^ PFU/mL) and 3.5% (1.88 × 10^3^ PFU/mL) compared with the control without compounds (5.42 × 10^4^ PFU/mL). Infection levels in the DENV-2/16681 model were 2.9% (5.00 × 10^2^ PFU/mL), 6.5% (1.13 × 10^3^ PFU/mL) and 0.9% (1.50 × 10^2^ PFU/mL) compared with the control without compounds (1.74 × 10^4^ PFU/mL). Statistically significant inhibition of infection was observed for the positive control (suramin) against DENV-2/NG and DENV-2/16681 strains (4.01 × 10^3^ PFU/mL, 7.4%; and 1.36 × 10^3^ PFU/mL, 7.8%, respectively) (Figure 6).

### 2.6. Non-Synonymous Amino Acid Differences in the Envelope Proteins of DENV-2/NG and DENV-16681

Considering that VOAC showed a higher virucidal effect against the DENV-2/NG strain than against the DENV-2/16681 strain (infection: 88% vs. 26%), the E protein from each strain was sequenced. Following confirmation of identity, 40 transitions and 3 transversions leading to 43 nucleotide substitutions were identified. To further understand the relevance of these modifications at the amino acid level, 435 amino acids corresponding to the nucleotide sequence were analyzed. Two synonymous (Thr120Arg and Ser478Thr) and four non-synonymous (Ser112Gly, Ile124Asn, Lys126Glu and Ala454Thr) modifications were identified. Three of these changes indicated differences in the amino acid polarity of the DENV-2/NG strain compared with the DENV-2/16681 strain, whereas the fourth indicated a difference in side-chain charge. It should be noted that the modifications at positions 112, 124, 126 and 454 were exclusive to the DENV-2/NG strain under study (Table 1).

### 2.7. The Energy of Binding of Indole Alkaloids and the Structural E Protein of the DENV Serotypes Is Favorable via Molecular Docking

The prediction of the interaction between the DENV-2 envelope protein domain III (PDB ID: 3UZV) and the indole alkaloids resulted in free energy of binding values between −5.1 ± 0.06 and −6.3 ± 0.21 kcal/mol. VOAC binding energy was −6.0 ± 0.06 kcal/mol and presented two hydrogen bonds formed with the same amino acid (Phe337); the value was −5.3 ± 0.10 kcal/mol with two hydrogen bonds (Lys334 and Asn355) for VOAC-OH; the binding energy was −5.1 ± 0.06 kcal/mol, with one hydrogen bond (Phe337) for rupicoline and the binding energy was −6.2 ± 0.15 kcal/mol for OXO-VOAC with three hydrogen bonds (Phe337, Val354 and Asn355) (Figure 7, Table 2). Three of the four isolated indole alkaloids exhibited hydrogen bonding with the amino acid Phe337. Despite this, the formation of this interaction in all cases was different due to the structural differences of the compounds. In the case of VOAC, the nitrogen of the indole group was the acceptor in the formation of the hydrogen bond formed with the hydroxyl group of the amino acid; contrary to the above, in the case of rupicoline and OXO-VOAC, the hydrogen bond was formed between a carboxyl group of the compound and the amino group of the amino acid but, while in rupicoline it was with the carbonyl group located at carbon 7 of the compound, in the OXO-VOAC it was with the carbonyl group of carbon 3 of the compound (Figure 7). On the other hand, although all compounds have the ester group at carbon 16, only OXO-VOAC formed a hydrogen bond with it. These differences show that, despite having the same indole nucleus as a possible pharmacophore and despite interacting in the same pocket of the protein, the structural differences generate a different molecular coupling of the compound in each case (Figure 7). Moreover, based on the virucidal potential of VOAC, its docking with the E proteins of other serotypes was assessed. Favorable energy of binding values was obtained for DENV-1, DENV-3 and DENV-4, with values of −5.9 ± 0.05, −5.6 ± 0.00 and −6.3 ± 0.05 kcal/mol, respectively. DENV-1 formed one hydrogen bond with Arg350, and DENV-3 formed two hydrogen bonds with Phe615 (Appendix A, Table 2).

### 2.8. The Interaction between DENV E Protein Domain III and VOAC Is Stable over Time via Molecular Dynamics

Molecular dynamics was conducted to detect the best interactions complex between compounds with virucidal effect and the domain IIII envelope protein (DIII-E) of inhibited DENV serotypes in the Trans-treatment strategy. The simulation between VOAC and DIII-E DENV-2 had an oscillation no greater than 0.25 nm (2.5 Å) during the time assessed (50 ns). The complexes formed between the same DIII-E DENV-2 and VOAC-OH, rupicoline and OXO-VOAC, had oscillations of up to about 4 nm (40 Å). The evaluation of the segments with the smallest oscillations of these complexes shows that in the case of VOAC-OH, the RMSD values ranged from 1.35 to 1.85 nm; rupicoline, 1 to 5 nm; and OXO-VOAC, 2.6 to 3.4 nm, approximately; at the times of 25 to 30 ns, 35 to 43 ns and 10 to 30 ns, respectively. The RMSD values of the complex DIII-E DENV-1 and VOAC values were up to 5 nm. The smallest oscillations were approximately 0.4 to 0.8 nm and occurred between 0 and 10 ns of the simulation. Therefore, the only stable complex was the one formed with voacangine and DIII-E DENV-2 (Figure 8).

## 3. Discussion

Despite the major public health problem that DENV infections represent, there is still no specific and effective antiviral therapy suitable for use in humans. For this reason, the search for such a therapy continues to be of vital importance. VOAC from *T. cymosa* plants has been recently shown to inhibit DENV-2 replication; thus, we proposed to identify other compounds of the same plant and further evaluate their antiviral potential using different approaches in various virus strains. Three indole alkaloids were isolated and identified, namely VOAC-OH, rupicoline and OXO-VOAC (Figure 1). This result was not surprising considering that the family Apocynaceae is known to be one of the plant families richest in alkaloids [20]. Moreover, several compounds of this type have been previously isolated from the species *T. cymosa* [29]. What differentiates the first and the third compounds structurally is the addition of a hydroxyl group and a ketone group, respectively, to VOAC (a major compound in the plant); in the second isolated compound, there is the oxidation of the seven-sided ring. However, it remains unclear if the identified modifications occur spontaneously in plants or if there are enzymes involved [30].

Cytotoxicity was dependent on the compound and its concentration when the viability of the Vero cell line treated with the three compounds was assessed (Figure 1D–F). This could be explained by the minor structural differences of each compound, which could induce differential toxicity, as has been previously reported [31,32]. In agreement with our findings, cytotoxicity has also been described as dependent on the concentration of indole alkaloids extracted from other *Tabernaemontana* species such as *T. inconspicua* [33]. This means that this type of compound has a wide range of applicability, from antimicrobials (at low dose) [34] to antineoplastic agents (at high dose) [35].

Taking into account that the antiviral effect of some compounds on serotype 2 depends on the virus strain and that both replication and protein expression differ in Vero cells according to the DENV-2 strain [8], we first carried out with the newly isolated compounds, and VOAC previously reported [17], four experimental approaches with DENV-2/NG and DENV-2/16681 strains. In the Pre-treatment, which was related to inhibition in the early stages of infection or with the modulation of cellular processes, inhibition was observed only in cultures first treated with VOAC-OH or rupicoline and then infected with the DENV-2/NG strain (Figure 2); whereas no compound presented antiviral activity in cultures infected with DENV-2/16681. Vero cells infected with these strains have been previously described as having differential cell protein expression [8]. Examples of this are the disulfide isomerase involved in the virus entry process [36], the calreticulin chaperone required for the production of infectious viral particles [37] and the proteasome subunit alpha that would be involved in the virus release [38], among others associated with the cellular metabolic processes. These differences in the proteins involved in the early and late stages of infection could be related to the strain-dependent activity observed for VOAC-OH and rupicoline. In addition to the above, it has been reported that the antiviral mechanism of arbidol, another compound with an indole core, is based on the inhibition of the fusion of the viral envelope with the endosomal membrane during the virus entry in the flavivirus hepatitis C virus model [39]. Therefore, this could also be considered a putative mechanism for VOAC-OH and rupicoline. In accordance with our results, previous studies have reported that VOAC showed no activity against DENV-2/NG in the Pre-treatment [17].

In the Trans-treatment, VOAC, VOAC-OH and rupicoline showed virucidal effects against both strains (DENV-2/NG and DENV-2/16681), whereas OXO-VOAC inhibited the DENV-2/16681 strain. Out of these three compounds, VOAC was the best virucide, with 88.8% inhibition of infection by DENV-2/NG (Figure 3). In addition, non-synonymous substitutions in the E protein of the NG and 16,681 strains were evinced by sequencing, where not only the amino acid but also its nature was changed (Table 1). Considering that the DENV E protein is essential for the interaction with cell receptors, such as heparan sulfate [40], laminin LAMR1 [41] and DC-SIGN [42], during the adhesion and virus entry processes, these substitutions could be associated with differences in virucidal activity among these compounds. These differences could be explained by possible structure reordering and steric hindrance due to the atomic radii of radical amino acid chains or by differences in charges that could directly affect docking and interaction of alkaloids and envelope proteins, thus acting as a virucide. Considering that VOAC was the best virucide, this compound was further assessed in strains belonging to the four DENV serotypes. VOAC showed activity only against DENV-1/Westpac/74 and DENV-2/S16803 (Figure 4). Similar results have been previously reported in cultures treated with a sulfated polysaccharide from the seaweed *Cladosiphon okamuranus,* which was capable of inhibiting serotype 2 but not the others [43]. Considering that a single mutation in the E protein can result in altered virus antigenicity, stability and pathogenesis [44], it could be stated that the virucidal effect would be associated with those modifications in the E protein of the different serotypes that affect the docking of the protein and the compound, thus explaining the dependence of the virucidal response on the serotype.

In contrast, no inhibition was observed for any of the compounds assessed in the Post-treatment (Figure 5). This was an unexpected result because it has been previously reported that VOAC inhibited DENV-2/NG in the post-treatment [17]. It could be hypothesized that this disagreement is due to the methodology used in each study; inhibition of DENV-2 replication has been determined by reverse transcription-quantitative polymerase chain reaction, but not by the decrease in infectious viral particles via plaque assay. In contrast to our results with the four alkaloids for the same treatment approach, an additional indole alkaloid such as hirsutine inhibited DENV infection [45]. This could be due to structural differences among the compounds despite sharing the same indole core.

In the combined strategy, three of the four indole alkaloids (VOAC, VOAC-OH and rupicoline) inhibited the infection caused by the DENV-2/NG and DENV-2/16681 strains. In all cases, inhibition percentages were higher than those obtained in individual approaches (Figure 6), which could indicate a synergic effect targeted to viral particles, viral proteins, and modulation of cellular processes promoting the antiviral effect, and therefore, this activity would be due to the constant presence of the compounds. Previous studies have described the promotion of antiviral activity in DENV-2 when comparing a combined approach using a compound such as orlistat with individual approaches [46]. Similarly, in another flavivirus model (ZIKV/Col with derivatives of the amino acid L-tyrosine), the same result of improved inhibition via combined approaches compared with individual ones was observed [47].

In silico tools, such as molecular docking and dynamics, have become an increasingly important support for drug discovery [48], as well as for modeling interactions of one molecule and one protein at the atomic level, thus allowing the characterization of the behavior in terms of ligand position and orientation in the interaction pocket, and the assessment of binding affinity [49]. In this regard, in silico docking was evaluated for the four indole alkaloids and the DENV E protein domain III. The compounds with virucidal activity against the two strains had the least favorable energy of binding within the studied group. This indicates that although in silico strategies are useful to predict protein–ligand interactions [50], it is necessary that in vitro assays accompany these observations and support the conclusions in an experimental context, to be able, in turn, to correlate the different strategies and give a biological meaning to the possible antiviral mechanism [51]. By contrast, favorable energy of binding was observed for VOAC with the E protein of all the serotypes, including those that were not inhibited by this compound (Table 2), reinforcing the need to accompany in silico approaches with in vitro assays.

When VOAC, the compound that presented the highest activity in the Trans-treatment against three DENV-2 strains, molecular docking was assessed, and hydrogen bond formation was observed with distances above 3.1 Å. Such distances have been described in stronger interactions, whereas distances up to 3.4 Å have been associated with weaker interactions [52]. Therefore, the VOAC-E protein interaction is mainly due to hydrophobic interactions (Figure 7). In addition, molecular dynamics showed that the only stable complex with E protein during 50 ns was VOAC (Figure 8), according to previous studies [53] that proves the stability of this complex. Furthermore, VOAC was the compound with the best virucidal activity demonstrated in three strains of DENV-2 (Figure 3 and Figure 4) this being the most interesting relationship between the in silico and in vitro results of the study, since the computational support, especially simulations, has become an important decision-making tool in antiviral research of promising compounds [51].

To conclude, we described the virucidal effect of VOAC isolated from *T. cymosa* against three different strains of DENV-2 (DENV-2/NG, DENV-2/16681 and DENV-2/S16803) and one strain of DENV-1 (DENV-1/WestPac/74), with there being a larger effect in cultures infected with DENV-2/NG. Moreover, the Combined treatment showed the best inhibitions of VOAC, VOAC-OH and rupicoline with effects independent of the DENV-2/strain. Therefore, taking into account these results, we could say that these compounds have an anti-DENV potential and that small changes in the chemical structure could improve or cancel the antiviral activity. As a limited natural source, studies with synthetic molecules and other analogs could be carried out in the future, in addition to determining the specific mechanisms of action of these compounds.

## 4. Materials and Methods

### 4.1. Isolation and Identification of Compounds

Seeds of T. cymosa plants were collected from the herbarium of the Botanical Garden Guillermo Piñeres (Cartagena, Colombia; Voucher No. JBC 3243). The plant material was exhaustively macerated with 90% ethanol, and this was corroborated by thin-layer chromatography (TLC). The solid/solvent ratio was 1:4 (*w*/*v*), and the resulting extract (15 g) was filtered and concentrated in a rotary evaporator. The compounds were then separated by open column chromatography (5 × 60 cm), using 200 g silica gel (Merck^®^, Darmstadt, Germany, 70–230 mesh) suspended in dichloromethane as the stationary phase. The extract was eluted using a gradient of increasing polarity starting with 100% dichloromethane, followed by 7:3 dichloromethane/acetone, 1:1 acetone/methanol, and 100% methanol. The purity of alkaloids was verified by measuring the melting points by TLC in three different solvent systems and by high-performance liquid chromatography (HPLC). HPLC was performed using the following conditions: Waters 1515 Isocratic HPLC Pump equipped with a UV-VIS detector (Waters 2489); Normal phase Column 5 µ SiGel PhenosphereNext 120Å, 250 × 4.6 mm with a 1 mL/min Flux. Mobil phase: Ethyl acetate. ^1^H and ^13^C nuclear magnetic resonance (NMR) were recorded at 300 and 75 MHz, respectively, with tetramethylsilane used as the internal standard. Units for chemical shifts were expressed in ppm. For column chromatography, silica gel 60 (Merck^®^, 70–230 mesh) was used. Prep thin-layer chromatography was performed on glass plates 20 × 20 × 0.1 cm on silica gel. Mass spectra at 70 eV were recorded. The compounds extracted from *T. cymosa* are subject to contracts for access to genetic resources and derived products #130 of 2016 (RGE0176) and #292 of 2020 (RGE0343) signed with the Ministry of Environment and Sustainable Development of the Republic of Colombia.

### 4.2. Virus and Cell Maintenance

C6/36 cells were donated by Dr. Raquel Ocazionez (Industrial University of Santander, Bucaramanga, Colombia) and cultured in L-15 medium supplemented with 10% FBS and 20 mM HEPES and kept at 28 °C. Vero epithelial cells (ATCC) were donated by Dr. Jorge Osorio, Department of Pathobiological Sciences, University of Wisconsin (Madison, WI, USA) and cultured in Dulbecco’s Modified Eagle’s Medium (DMEM) supplemented with 10% SFB and 1% antibiotic/antifungal (10 mg/mL streptomycin, 10,000 U/mL penicillin and 0.025 mg/mL amphotericin B, GIBCO^®^, Grand Island, NY, USA), and kept at 37 °C in a humidified atmosphere containing 5% CO_2_. Six virus reference strains were used, namely DENV-2/NG, DENV-2/16681, DENV-1/Westpac/74, DENV-2/S16803, DENV-3/16562 and DENV-4/Indonesia-1976.

### 4.3. Determination of the Cytotoxicity of Isolated Compounds

Cytotoxicity was assessed using the 3-(4,5-dimethylthiazole-2-yl)-2,5-diphenyltetrazolium bromide (MTT) (Sigma-Aldrich, St. Louis, MO, USA) assay. Cells were seeded (96 wells, 3 × 10^4^ cells/well), and serial dilutions of each compound were added after 24 h. After an incubation of 48 h, the compounds were removed, and 0.5 mg/mL MTT was added. DMSO was added 2 h later. Absorbance was measured at 450 nm using a microplate reader (Multiskan™ FC Microplate Photometer, Thermo Scientific, Waltham, MA, USA). In every case, cultures without the addition of compounds were used to represent 100% viability. Each assay was performed in two independent experimental units with three replicates each (*n*: 6).

### 4.4. Antiviral In Vitro Assays

#### 4.4.1. Antiviral Strategies

Four approaches were used to assess the antiviral effect. The Pre-treatment was used to evaluate the effect of the compounds on the cells before infection. For this purpose, 6.0 × 10^4^ cells were seeded in 48 wells, and subsequently, each compound was added at a non-cytotoxic concentration. After a 48 h incubation, the supernatant was removed, and the monolayers were inoculated with each virus strain (MOI: 1) for 2 h. Plates were incubated for 48 h. The Trans-treatment sought to assess the virucidal effect. For this purpose, 6.0 × 10^4^ cells were seeded in 48 wells, and after 24 h, viral inoculum (MOI: 1) and compounds at a non-cytotoxic concentration were combined in equal parts. This combination was incubated at 4 °C for 1 h. The cell medium was then removed, and the combination was added onto the monolayers and incubated at 37 °C for 2 h. The viral inoculum was then removed, and fresh media was added for further incubation of 48 h. The Post-treatment was used to evaluate the antiviral effect of the compounds on the stages following the entry of the virus into the cell. In this case, 6.0 × 10^4^ cells (48 wells) or 3.0 × 10^4^ cells (96 wells) were seeded. After a 24 h incubation, the cell media was removed, and the viral inoculum (MOI: 1) added for 2 h. The inoculum was then taken out, and the compounds were added and incubated for 48 h. Finally, to assess the three previous approaches successively, a Combined treatment was used. The seeded cells were first incubated with the compounds. The compounds were then removed, and the cells inoculated for 2 h with the combination virus/compound (1:1). The inoculum was then taken out, and a fresh compound added and incubated for 24 h. For all the approaches, controls without compounds (CWC) (100% infection) and positive inhibition controls were used. After inoculation, supernatants were collected and kept at −80 °C until plaque titration. Each assay was performed in two independent experimental units with two replicates each (*n*: 4).

#### 4.4.2. Quantification of Infectious Viral Particles by Plaque Assay

Vero cells (6 × 10^4^) were seeded in 48 wells and inoculated after 24 h with serial dilutions of the supernatants. After a 2 h incubation, the viral inoculum was removed, and 1 mL 1.5% carboxymethyl cellulose (Sigma-Aldrich, St. Louis, MO, USA) was added to DMEM supplemented with 2% FBS (Gibco, Grand Island, NY, USA). After 12 days, monolayers were fixed using 4% paraformaldehyde (Sigma-Aldrich, St. Louis, MO, USA), stained with crystal violet, and plaques were counted.

#### 4.4.3. Statistical Analysis

The normality of the data was determined using the Shapiro–Wilk to confirm normality. The number of infectious viral particles released between treated and untreated cells under the different experimental conditions were compared. Statistically significant differences were identified using a parametric Student’s *t*-test (data with normal distribution). Statistical analyses were performed with the Prism^®^ 7.01 package for Windows™ (GraphPad Software, San Diego, CA, USA). In every case, *p* values lower than 0.05 (*p* < 0.05) were considered statistically significant.

### 4.5. In Silico Studies

#### 4.5.1. Gene Sequence Analysis

For the DENV-2 strains used in this study, E protein was amplified using SuperScript™ III One-Step RT-PCR System with Platinum™ Taq DNA Polymerase (Thermo Fisher Scientific, Waltham, MA, USA), according to the manufacturer’s instructions and previously reported primer sequences and protocols [54]. Amplicons were sequenced using the Sanger method (Macrogen Inc., Seoul, Korea). Sequences were assembled and edited with the Lasergene software, and a BLAST search was performed using the online platform Dengue Virus Typing Tool v.3.82. Subsequently, reference sequences retrieved from NCBI and the sequences under study were aligned with CLUSTAL W using MEGA v7. To determine nucleotide differences between strains, 1308 nucleotides corresponding to 435 amino acids of the E protein of the DENV-2 New Guinea C (AB609589) and DENV-2 16,681 (NC001474) strains were compared with those of the strains assessed here (DENV-2/NG and DENV-2/16681, respectively).

#### 4.5.2. Selection of the Target Protein

The three-dimensional structure of Envelope proteins with resolution lower than 3.0 Å was downloaded from the Protein Data Bank (PDB) database. In silico computational analysis was performed for each E protein of DENV 1–4 serotypes with PDB IDs 4FFY, 3UZV, 3VTT and 3WE1, respectively [55]. Structures were prepared using Python Molecular Viewer (PMV) software to initially remove water molecules from the system; polar hydrogen atoms were then added to favor possible hydrogen bonds; Gasteiger–Marsili charges were calculated, thus minimizing the system’s energy. The minimized structure was saved as a PDBQT file for docking studies.

#### 4.5.3. Selection of Ligand

The structures of the three indole alkaloids were retrieved from the PubChem database (https://pubchem.ncbi.nlm.nih.gov/, accessed on 8 May 2018); VOAC (ID: 10361692), VOAC-OH (ID: 328232), and rupicoline (ID: 101593056). The structure of OXO-VOAC was modeled using ACD/ChemSketch ^®^ 12.01 software (Freeware Version, Toronto, ON, Canada). Ligand flexibility was configured with the PMV tool that specifies ligand torsions. The file was saved as PDBQT for docking studies.

#### 4.5.4. Molecular Docking

Target protein–ligand interactions were analyzed in triplicate using AutoDock Vina software [56]. The best binding site or hot spot on the surface of the target protein was identified with the online tool PeptiMap [57]. Based on the predicted sites, grid box dimensions X-Y-Z were defined as 30, and the exhaustiveness value was set at 10. The best free energies of binding in terms of kcal/mol that met the acceptance criterion of between 0 and −16 were selected. In addition, intermolecular interactions (hydrogen bonds and hydrophobic interactions) were analyzed by two-dimensional diagrams using LigPlot software (https://www.ebi.ac.uk/thornton-srv/software/LIGPLOT/, accessed on 18 March 2019).

#### 4.5.5. Molecular Dynamics

Molecular dynamics were conducted to detect the best interaction complexes between structural proteins (domain III of DENV-2 and DENV-1 envelope protein) and virucidal compounds. These simulations were assessed using the GROMACS^®^ software (Groningen MAchine for Chemical Simulation, developed at the University of Groningen, The Netherlands. Accessed on September 08, 2020) [58] based on the previous theorists of simulation for the protein–ligand complex [59]. The topology of target protein and ligand were obtained, and the force field for both components was generated by CHARMM software [60] and the official CHARMM General Force Field (CGenFF) server [61], respectively. The topology of the protein–ligand of each complex was constructed, and the conditions of the water box were established in a neutral ionic environment. The system was brought to minimum energy and was balanced. The simulations were carried out for 50 ns with the number of molecules, temperature and pressure (NVT and TPN) constant [60,62]. The trajectories and a plot of distances in the time assessed were obtained. RMSD plots of the in silico simulation were analyzed, considering oscillations of less than 0.3 nm (3 Å) as stable or biologically feasible complexes [63,64].

## Figures and Tables

**Figure 1 plants-10-01280-f001:**
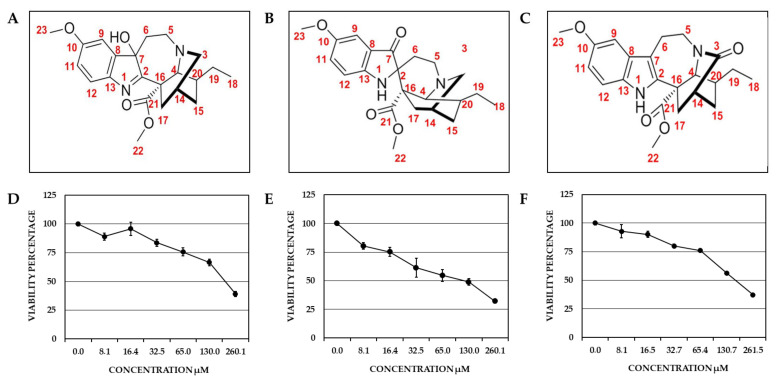
Chemical characterization and cytotoxicity of isolated compounds from *T. cymosa.* Three compounds were identified, and all were structurally related with VOAC [17]; VOAC-OH (**A**); Rupicoline, also called voacangine pseudo-indoxyl (**B**); and OXO-VOAC (**C**). The viability of these compounds was determined by the MTT method at 2:1 serial dilution (*n*: 6). For VOAC-OH, concentrations from 8.1 to 260.1 µM (**D**); Rupicoline, from 8.1 to 260.1 µM (**E**) and OXO-VOAC, from 8.1 to 261.5 µM (**F**) were evaluated.

**Figure 2 plants-10-01280-f002:**
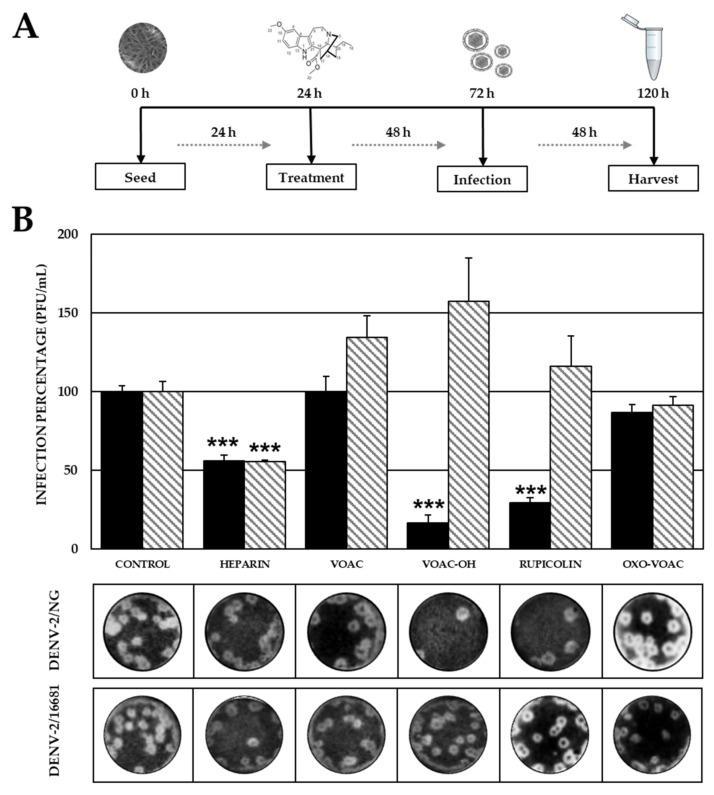
Antiviral effects on viral production (DENV-2) using the Pre-treatment strategy. (**A**) Schematic representation of the Pre-treatment strategy explained in the Materials and Methods section. (**B**) Infection percentage in each experimental condition. Vero cells were pre-treated for 48 h and then infected with DENV-2/NG (Dark bars) or DENV-2/16681 (Clear bars). The asterisks indicate statistically significant differences with respect to the control without compound (*** *p* < 0.001; *t*-Student) and error bars indicate standard error of the mean; *n*: 4. Moreover, it shows representative images of plaque formed in each experimental condition.

**Figure 3 plants-10-01280-f003:**
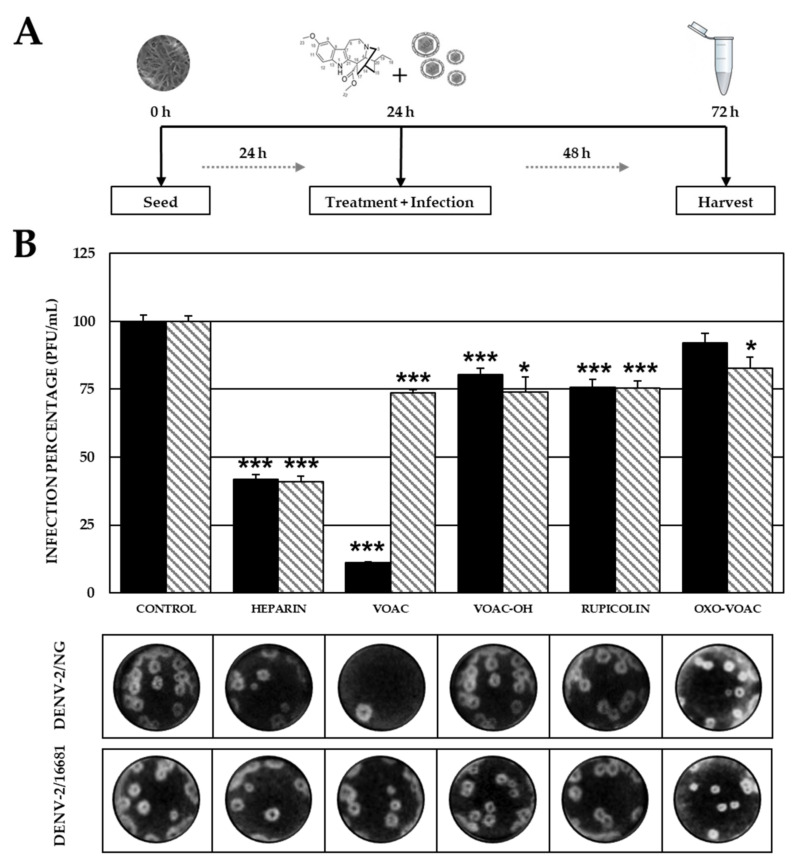
Virucidal effect against DENV-2 strains. (**A**) Schematic representation of the Trans-treatment strategy explained in the Materials and Methods section. (**B**) Infection percentage in each experimental condition. Vero cells were treated and infected at the same time with DENV-2/NG (Dark bars) or DENV-2/16681 (Clear bars). The asterisks indicate statistically significant differences with respect to the control without compound (* *p* < 0.05 and *** *p* < 0.001; *t*-Student) and error bars indicate standard error of the mean; *n*: 4. Moreover, it shows representative images of plaque formed in each experimental condition.

**Figure 4 plants-10-01280-f004:**
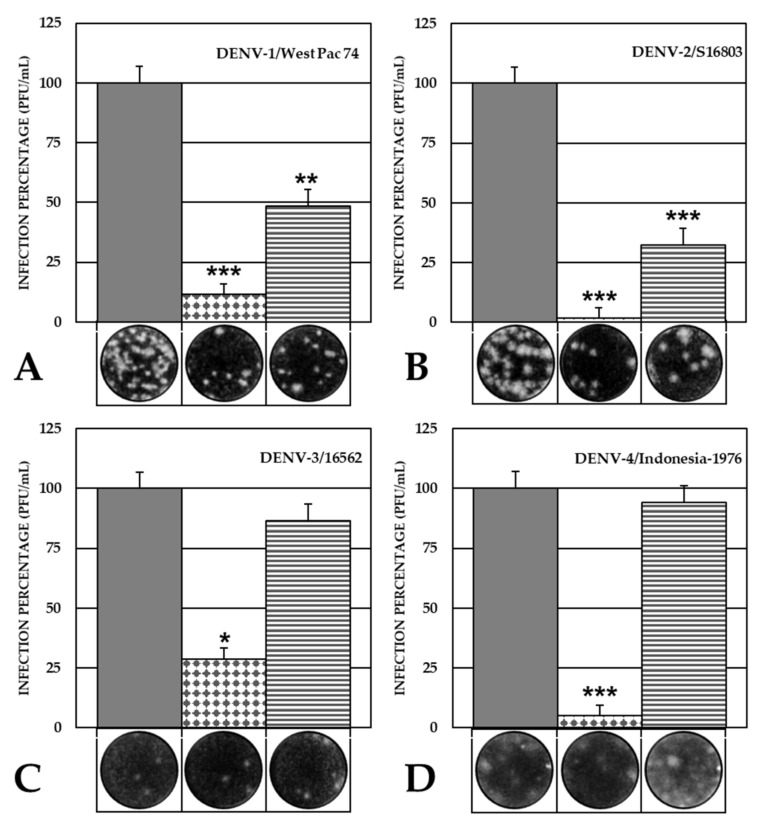
Virucidal effect against DENV serotypes. Antiviral effect of VOAC using the Trans-treatment strategy against standard strains of each one of the four DENV serotypes. For all the approaches, grey bars indicate controls without compounds (100% infection), dotted bars indicate the inhibition control (Heparin) and striped bars indicate VOAC treatment. Cell culture were infected with DENV-1/WestPac/74 (**A**), DENV-2/S16803 (**B**), DENV-3/16562 (**C**) and DENV-4/Indonesia-1976 (**D**). The asterisks indicate statistically significant differences with respect to the control without compound (* *p* < 0.05, ** *p* < 0.01 and *** *p* < 0.001; *t*-Student) and error bars indicate standard error of the mean; *n:* 4. Moreover, it shows representative images of plaque formed in each experimental condition.

**Figure 5 plants-10-01280-f005:**
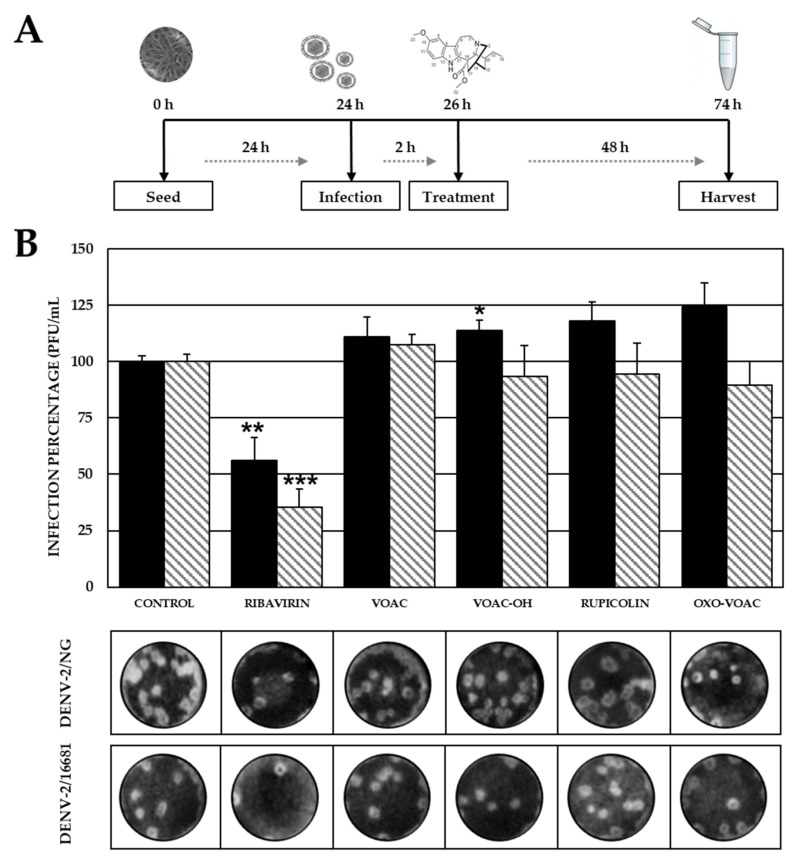
Antiviral effects on viral production (DENV-2) using the post-treatment strategy. (**A**) Schematic representation of the post-treatment strategy explained in the Materials and Methods section. (**B**) Infection percentage in each experimental condition. Vero cells were infected with DENV-2/NG (Dark bars) or DENV-2/16681 (Clear bars) and after that were treated. Infection percentage in each experimental condition. The asterisks indicate statistically significant differences with respect to the control without compound (* *p* < 0.05, ** *p* < 0.01 and *** *p* < 0.001; *t*-Student) and error bars indicate standard error of the mean; *n*: 4. Moreover, it shows representative images of plaque formed in each experimental condition.

**Figure 6 plants-10-01280-f006:**
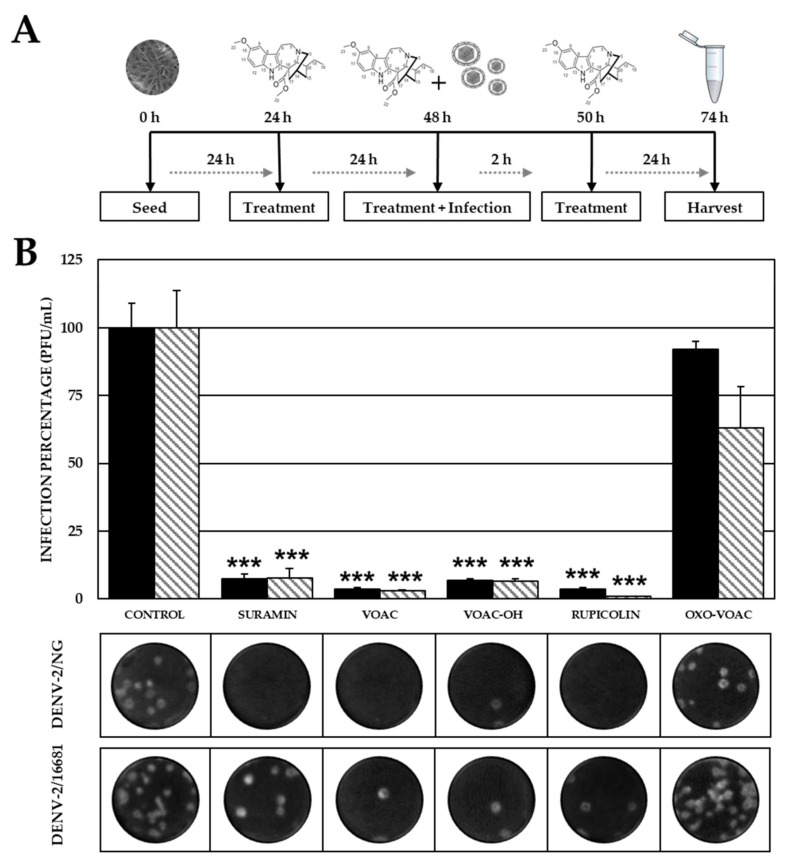
Anti-DENV-2 activity with the Combined treatment strategy. (**A**) Schematic representation of the combinate treatment strategy explained in the Materials and Methods section. (**B**) Infection percentage in each experimental condition. Vero cells were treated with each one of the compounds previous, during and after the infection with DENV-2/NG (Dark bars) or DENV-2/16681 (Clear bars). The asterisks indicate statistically significant differences with respect to the control without compound (*** *p* < 0.001; *t*-Student) and error bars indicate standard error of the mean; *n*: 4. Moreover, it shows representative images of plaque formed in each experimental condition.

**Figure 7 plants-10-01280-f007:**
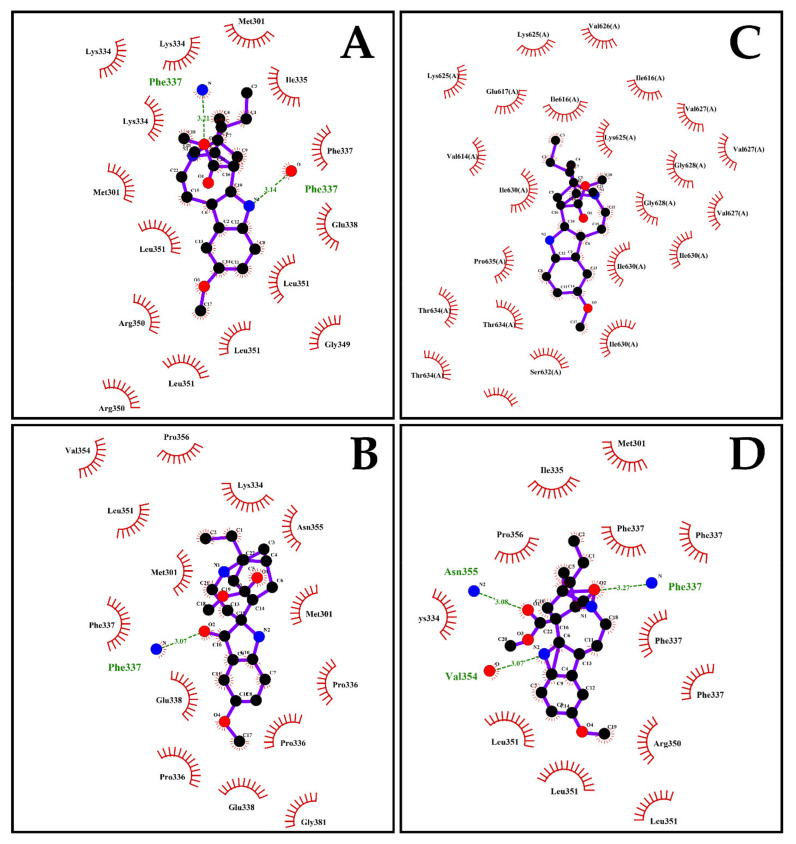
Molecular docking between compounds and DENV-2 envelope protein. The interactions formed between Domain III of DENV-2 Envelope protein and VOAC (**A**), VOAC-OH (**B**), rupicoline (**C**) and OXO-VOAC (**D**) were evaluated by LigPlot^®^. Hydrogen bonds are shown in green and hydrophobic interactions in red.

**Figure 8 plants-10-01280-f008:**
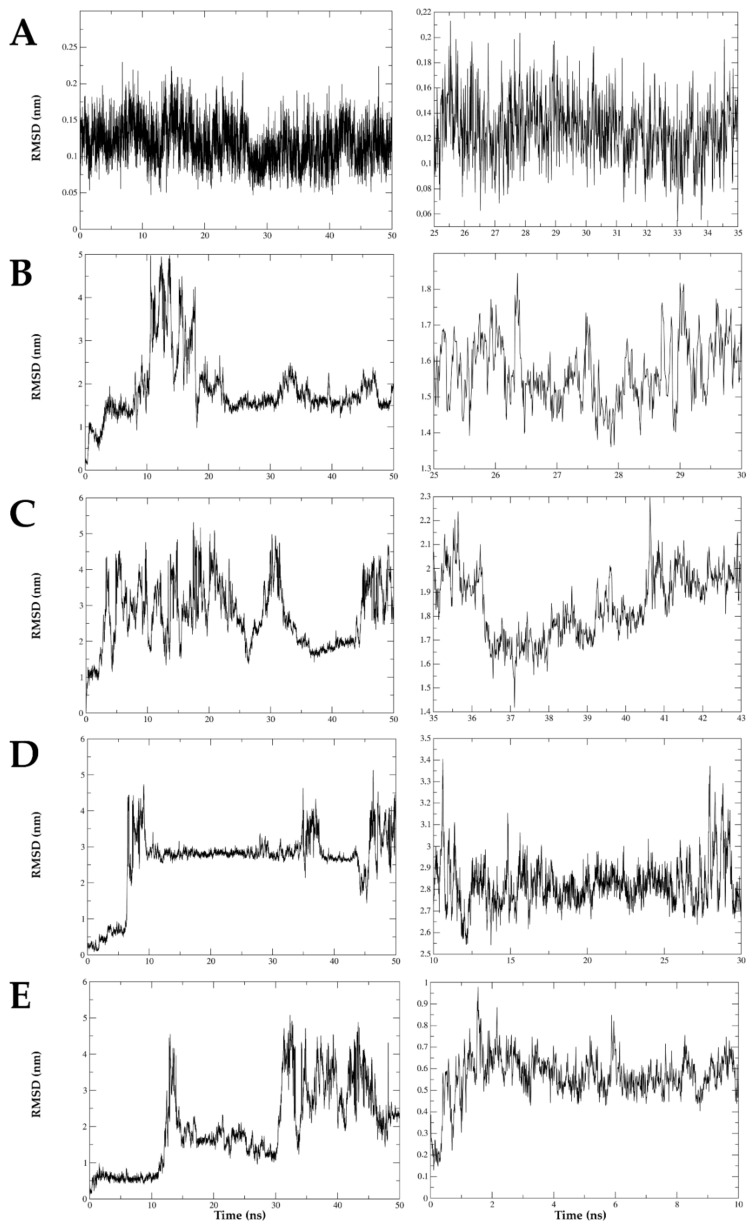
Complex stability by molecular dynamics. The stability of the complex formed by each compound with virucidal effect and the Domain III of DENV envelope protein was evaluated; DENV-2 with VOAC (**A**), VOAC-OH (**B**), rupicoline (**C**) and OXO-VOAC (**D**), and DENV-1 with VOAC (**E**). The left plot indicates the complete 50 ns simulation, and the right plot indicates the site with the lowest oscillation of each simulation. The *y*-axis represents the root-mean-square deviation (RMSD) that represents the average oscillation of the distance between the atoms of the complex components in nanometers (nm), and the *x*-axis the timescale in nanoseconds (ns). The complex is considered stable if the oscillation from the initial position of the complex is below 0.3 nm.

**Table 1 plants-10-01280-t001:** Amino acid changes in the envelope gene of the DENV-2/NG and DENV-2/16681 strains.

		Aligned Sequence					
	Position	112	120	124	126	454	478
Reference sequence	AB609589 DENV-2 New Guinea C	G	T	N	E	T	S
Target sequence	DENV-2/NG (Sample 1)	S	T	I	K	A	S
Reference sequence	NC001474 DENV-2 16681	G	R	N	E	T	T
Target sequence	DENV-2/16681 (Sample 2)	G	R	N	E	T	T

**Table 2 plants-10-01280-t002:** Molecular docking and in silico interactions between DENV and each of the compounds.

Ligand	Serotype	Free Binding Energy (Kcal/mol)	Hydrogen Bonds	Residues Forming H Bonds	Distance between H+ Bonds (Å)	Residues Participating in Hydrophobic Interactions
VOAC	DENV-1	−5.90 ± 0.05	2	Arg350	3.22	Lys334-Thr339-Phe337-Ser338-Arg350-Gly349-Pro371-Glu370-Leu351
	DENV-2	−6.03 ± 0.06	1	Phe337 (×2)	3.21−3.14	Met301 (×2)-Lys334 (×3)-Ile335-Phe337-Glu338-Leu (×4)-Arg350(×2)-Gly349
	DENV-3	−5.60 ± 0.00	2	Phe615	2.99−3.10	Leu629-Lys612-Met579-Arg628-Gly627-Pro614-Ile613-Phe615
	DENV-4	−6.30 ± 0.05	N/A	N/A	N/A	Ile616-Val626-Lys625-Val627-Gly628-Ile630-Ser632-Thr634-Pro635-Val614-Glu617
VOAC-OH	DENV-2	−5.30 ± 0.10	2	Lys334−Asn355	3.20−3.25	Lys334 (×2)-Met301-Phe337-Ile335-Pro356-Leu351 (×2)-Val354
Rupicoline	DENV-2	−5.07 ± 0.06	1	Phe337	3.07	Pro356-Pro336(×3)-Val354-Leu351-Lys334-Asn355-Met301 (×2)-Phe337-Glu338 (×2)-Gly381
OXO-VOAC	DENV-2	−6.17 ± 0.15	3	Asn355−Val354 −Phe337	3.08−3.07−3.27	Met301-Ile335-Pro356-Phe337 (×4)-Lys334-Leu351 (×3)-Arg350

## Data Availability

The data presented in this study are available on reasonable request from the corresponding author.

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
