# Peer review of "The Antiviral and Virucidal Activities of Voacangine and Structural Analogs Extracted from Tabernaemontana cymosa Depend on the Dengue Virus Strain"

_plants, 2021, doi:10.3390/plants10071280_

Round 1
Reviewer 1 Report
The paper describes the effects of four alkaloids obtained from T. cymosa seeds over dengue virus strains. Interesting results were obtained such as the antiviral effect dependent on the serotype and the virus strain.
I suggest three minor revisions on the paper:
Lines 137, 147 and 158: These spectra were compared .....
From my point of view, in Figure 1 the NMR spectra are unnecessary. The size is small and due to it any interpretation is very difficult. These spectra could be inserted in the Supplementary material.
The quality of Figures 7 & 8 must be improved.
Author Response
REVIEWER 1
The paper describes the effects of four alkaloids obtained from T. cymosa seeds over dengue virus strains. Interesting results were obtained such as the antiviral effect dependent on the serotype and the virus strain. I suggest three minor revisions on the paper:
Specific comments:
- Lines 137, 147 and 158: These spectrawere compared
Answer from author:
The word “specter” was changed to spectra according to the reviewer comment (Line 141, 151 and 161)
- From my point of view, in Figure 1 the NMR spectra are unnecessary. The size is small and due to it any interpretation is very difficult. These spectra could be inserted in the Supplementary material.
Answer from author:
We agreed with the reviewer comment and we moved the NMR spectra to Supplementary material (Figure S1, S2 and S3). Moreover, we included the new description of those figures in section supplementary materials (Lines 636-639) and modify the figure 1 and its legend (177-181)
- The quality of Figures 7 & 8 must be improved.
Answer from author:
The quality of figures 7 and 8 was improved and the files were uploaded in the submission system. Moreover, to ensures that our figures meet the quality request we confirm through the PACE tool: https://pacev2.apexcovantage.com/
Reviewer 2 Report
It is good experimental article with interesting subject and good experimental work. The manuscript is fairly well written and includes a great deal of information, which is reflected in the great number of references listed.
Below please find some of my comments:
Abstract – every time abstract should contains the most important information like most important findings and results. Some values are needed. The abstract should be reorganized.
English and style require a careful reorganization. There are a lot of language mistakes, grammatically and stylistically.
Introduction section - Please cite references for each information presented in the Introduction section (lines 111-113).
The results and discussion are represented in a logical way.
Materials and Methods - How many repeats of maceration was performed? That question further raises another one, how many repeats were performed in further biological analysis? What was solid to liquid ratio?
The Conclusions must reflect the innovation of this study and the perspectives. The results should me more emphasized to interest readers in the subject.
Author Response
REVIEWER 2
It is good experimental article with interesting subject and good experimental work. The manuscript is fairly well written and includes a great deal of information, which is reflected in the great number of references listed.
Specific comments:
- Abstract– every time abstract should contains the most important information like most important findings and results. Some values are needed. The abstract should be reorganized.
Answer from author:
According to the reviewer comment, we reorganized the abstract, eliminating information not-relevant, and including the values of infection percentages to made emphasis in the most important findings (Line 27 – 29; 33, 35, 36, 37, 39 and 41-42).
- English and style require a careful reorganization. There are a lot of language mistakes, grammatically and stylistically.
Answer from author:
The paper was translated by ENAGO and, moreover, we check the language carefully across the manuscript. However, if the reviewer considers necessarily make a new English edition, we can ask to ENAGO by a guaranty of the translation service.
- Introductionsection - Please cite references for each information presented in the Introduction section (lines 111-113).
Answer from author:
According to the recommendation we include four new cite references that reported different biological activities of the genus Tabernaemontana (Pereira et al, 2003; Thambi et al, 2006; de Almeida et al, 2004 and Van Beek et al, 1984). Line 114-115.
- The results and discussion are represented in a logical way.
Answer from author:
Thanks for the comment. However, we tried to improve some specific items according to the recommendations of the other reviewer.
- Materials and Methods- How many repeats of maceration was performed?
Answer from author:
The maceration is carried out until the material is exhausted, that is, until it is evident that there are no metabolites present in the extraction solvent (which is corroborated by TLC). In the text the phrase "The plant material was macerated with 90% ethanol overnight" was changed by “The plant material was exhaustively macerated with 90% ethanol and this was corroborated by Thin Layer Chromatography (TLC)” (Line 510-516).
- That question further raises another one, how many repeats were performed in further biological analysis?
Answer from author:
The biological assays were made in different replicates. This information can be saw in section 4.3, Lines 541-542 (Each assay was performed in two independent experimental units with three replicates each (n: 6)) and section 4.4., Lines 566-568. Moreover, in each figure legend, the number of replicates was clarified (Line 196, 214, 230, 246 and 264).
- What was solid to liquid ratio?
Answer from author:
The Solid/liquid ratio was 1:4. This information was added in Line 504.
- The Conclusions must reflect the innovation of this study and the perspectives. The results should be more emphasized to interest readers in the subject.
Answer from author:
According to the reviewer recommendations, we improved the conclusion, emphasizing the most important results and the perspectives. Line 489-498.
Reviewer 3 Report
In the present study, three alkaloids were isolated from seeds of Tabernaemontana cymosa and their in vitro antiviral effect against different virus strains was assessed using different experimental approaches. In silico studies were also performed to explain the possible antiviral mechanism. As result, the PRE-treatment using voacangine hydroxyindolenine and rupicoline inhibited the infection caused by the DENV-2/NG strain. In the TRANS-treatment approach, voacangine, voacangine hydroxyindolenine, and rupicoline inhibited the infection in both DENV-2/NG and DENV-2/16681 infection models. The latter strain was also inhibited by 3-oxo-voacangine. Moreover, voacangine was also effective against one strain of DENV-1 (DENV-1/WestPac/74) and against the third strain of DENV-2 (DENV-2/S16803). In silico results showed that there are differences in the coding sequences of the virus strains used in the present study. The free energy of binding was favorable for the E protein and compounds, which ranged between −5.1 and −6.3 kcal/mol. Finally, the complex formed between DENV-2 E protein and the best antiviral compounds was stable for 50 ns. Obtained results evidenced that alkaloids derived from this plant are promising molecules for the development of anti-DENV drugs and that this effect depends on the serotype and the virus strain.
In the opinion of this referee, this work was well conducted and presents interesting and promising results concerning the in vitro antiviral effect against different Dengue virus strains. Therefore, I suggest that this manuscript could be accepted to publication in Plants after minor revision, as follow:
1. Considering the future use of these related alkaloids as antiviral compounds, did the authors perform some studies in order to evaluate the cytotoxicity against mammalian cells (NCTC for example)?
2. The authors showed the NMR spectra of each isolated compound – in my opinion, this is unnecessary and could be transferred to supplementary material. However, an important aspect was not discussed in the manuscript. How did the authors determine the purity of tested alkaloids? Is this performed using exclusively NMR or other methods (especially chromatographic) were used?
3. In combination with in silico studies, I suggest that the authors describe a comparison of the chemical structures of these related alkaloids and the observed biological potential. This approach could suggest aspects about the presence of some important pharmacophore groups in these compounds. This point must be included in the revised version of this manuscript.
Author Response
REVIEWER 3
In the present study, three alkaloids were isolated from seeds of Tabernaemontana cymosa and their in vitro antiviral effect against different virus strains was assessed using different experimental approaches. In silico studies were also performed to explain the possible antiviral mechanism. As result, the PRE-treatment using voacangine hydroxyindolenine and rupicoline inhibited the infection caused by the DENV-2/NG strain. In the TRANS-treatment approach, voacangine, voacangine hydroxyindolenine, and rupicoline inhibited the infection in both DENV-2/NG and DENV-2/16681 infection models. The latter strain was also inhibited by 3-oxo-voacangine. Moreover, voacangine was also effective against one strain of DENV-1 (DENV-1/WestPac/74) and against the third strain of DENV-2 (DENV-2/S16803). In silico results showed that there are differences in the coding sequences of the virus strains used in the present study. The free energy of binding was favorable for the E protein and compounds, which ranged between −5.1 and −6.3 kcal/mol. Finally, the complex formed between DENV-2 E protein and the best antiviral compounds was stable for 50 ns. Obtained results evidenced that alkaloids derived from this plant are promising molecules for the development of anti-DENV drugs and that this effect depends on the serotype and the virus strain.
In the opinion of this referee, this work was well conducted and presents interesting and promising results concerning the in vitro antiviral effect against different Dengue virus strains. Therefore, I suggest that this manuscript could be accepted to publication in Plants after minor revision, as follow:
Specific comments:
- Considering the future use of these related alkaloids as antiviral compounds, did the authors perform some studies in order to evaluate the cytotoxicity against mammalian cells (NCTC for example)?
Answer from author:
Taking in account, that Vero cells are the model more used to evaluate the anti-DENV effect, in this study we evaluate the cytotoxicity of alkaloids only in these cells. However, in other study of our team (in evaluation process in BMC Complementary Medicine and Therapies) we evaluated the cytotoxicity in mammalian cells (U937 and A549) as a model of infection for ZIKV and CHIKV and results demonstrated that the effect is very similar between different mammalian cells.
- The authors showed the NMR spectra of each isolated compound – in my opinion, this is unnecessary and could be transferred to supplementary material
Answer from author:
We agreed with the reviewer comment, and we moved the NMR spectra to Supplementary material (Figure S1, S2 and S3). Moreover, we included the new description of those figures in section supplementary materials (Lines 636-639) and modify the figure 1 and its legend (177-181)
- However, an important aspect was not discussed in the manuscript. How did the authors determine the purity of tested alkaloids?
Answer from author:
Purity of alkaloids was verified by measuring the melting points, by TLC in three different solvent systems and High-Performance Liquid Chromatography (HPLC). This information was included in Line 509-514.
- Is this performed using exclusively NMR or other methods (especially chromatographic) were used?
Answer from author:
We also measured the melting points and used analytical TLC and HPLC. HPLC was performed using the following conditions: Waters 1515 Isocratic HPLC Pump equipped with a UV-VIS detector (Waters 2489); Normal phase Column 5µ SiGel PhenosphereNext 120Å, 250 x 4.6 mm with a 1mL/min Flux. Mobil phase: Ethyl acetate. This information was included in section 4.1 (Line 509-514). Moreover, in section 2.1. We include this sentence “Reported alkaloids showed over 90% purity” (Line 132) and the melting point was included for each compound (Line 140-141; 150-151 and 60-161).
- In combination with in silico studies, I suggest that the authors describe a comparison of the chemical structures of these related alkaloids and the observed biological potential. This approach could suggest aspects about the presence of some important pharmacophore groups in these compounds. This point must be included in the revised version of this manuscript.
Answer from author:
According to the reviewer suggestion we include in section 2.7 a paragraph that included an explanation about this topic (Line 291-304).
This manuscript is a resubmission of an earlier submission. The following is a list of the peer review reports and author responses from that submission.
Round 1
Reviewer 1 Report
The article by Marlen Martinez-Gutierrez entitled "The antiviral and virucidal activities of voacangine and structural analogs extracted from Tabernaemontana cymosa depend on the Dengue virus strain", in my opinion, does not warrant acceptance for publication in Molecules at this stage.
In the manuscript the authors described isolation of five indole alkaloids from Tabernaemontana cymose and evaluated their activities.
The compound data is significantly inadequate. The reason is as follows.
Compounds A, B, C, and D were found in scifinder. While, I could not find compound E in scifinder, which means it is a new compound. I do not think that the authors checked whether new or not. It is very important. If E is new, compound charactalization of E is significantly insufficient. The authors should submit the following data; copies of 1H and 13C NMRs, HRMS, IR, melting point (if solid). In 1H NMR, the authors should need the number of 1H (not position). Therefore, the results of subsequent assays are also unreliable.
After revision, I recommend that the manuscript will be submitted some journals which is focused on medicinal chemistry.
END
Reviewer 2 Report
The manuscript dealt with antiviral and virucidal activities of voacangine and structural analogs extracted from T. cymosa. The manuscript is well written and well referenced. The compounds of interest were however not clearly identified. For instance the amount (wt) of each compound is not stated; only 1H & 13C-nmr were reported for the compounds, an accurate mass or hplc purity will inform one of how pure the compounds are. I will like for the weights of the compounds and their accurate masses to be included.
Reviewer 3 Report
The manuscript is very interesting and is well-structured in general. The experimental panel was well-planned with the use of different techniques used in molecular sciences. In my opinion the obtained data could be interesting for a reasonable number of scientists since among the infections caused by viruses transmitted by arthropods, dengue fever is the most frequent. However the Authors should undertake revision which in my opinion would improve their study:
1.Results section - from the description of cytotoxic assay it is very hard to follow the obtained results. This part should be re-written regarding the viability data and the used cell lines. Moreover, according to the Figure 1 all results presenting the viability data should be demonstrated.
2. Materials and Methods section - 4.3. subsection - in the description of cytotoxicity the used cell lines and the apllied range of concentrations of the tested compounds should be added.
In my opinion, after these changes, the manuscript merit publication.